# Project Portfolio Planning Taking into Account the Effect of Loss of Competences of Project Team Members

Grzegorz Bocewicz [1,*], Eryk Szwarc [1], Amila Thibbotuwawa [2] and Zbigniew Banaszak [1]

1    Faculty of Electronics and Computer Science, Koszalin University of Technology, 75-453 Koszalin, Poland; eryk.szwarc@tu.koszalin.pl (E.S.); zbigniew.banaszak@tu.koszalin.pl (Z.B.)
2    Center for Supply Chain, Operations and Logistics Optimization, University of Moratuwa, Moratuwa 10400, Sri Lanka; amilat@uom.lk
*    Correspondence: grzegorz.bocewicz@tu.koszalin.pl

**Abstract:** This paper deals with a declarative model of the performance of employees conducting variably repetitive tasks based on the assumption of aging competences. An analytical model is used to consider refreshing the competences of the team's multi-skilled members and shaping the structure of staff's competences to maximize their mutual substitutability in processes typical for a multi-item lot-size production. Its impact on maintaining the skill level of employees is important in cases of an unplanned event, e.g., caused by employee absenteeism and/or a change in the priorities of orders carried out, disrupting the task of software companies. The developed model implemented in the constraint programming environment enables the formulation of decision-making versions of both the problem of analysis (seeking an answer to the question to discover whether there is a solution that meets the given expectations) and synthesis (seeking an answer to the question, assuming there is a solution that meets the given expectations). The potential of the proposed reference model-based approach is illustrated with examples.

**Keywords:** dynamic software multi-project scheduling; multitasking skills; competences' maintenance; job rotation





## 1. Introduction

The challenges posed by the next stages of the industrial revolution, in particular Industry 4.0 and Industry 5.0, highlight the need for a transition to a sustainable, human-centered and robust industry [1–3]. The drive towards human-centered production, which is visible, among other aspects, in the tendency to integrate a human-in-the loop concept with technologies, shows the emerging paradigm of a system focused on sustainable human development [4]. The expectations associated with this force the development of competence skills (i.e., the preparation of trained professionals who have competences and skills in new reprogrammable technologies), ensuring the diversity of employees in terms of experience, efficiency and physical abilities that guarantee the competitiveness of modern enterprises [5]. For example, support for systems such as SAP, ORACLE or JDA is required as core competences are expected from the ERP implementers' sector.

Unfortunately, the availability of multi-skilled employees necessary in the conditions of short-run or unit multi-assortment production [6,7], as well as the need to guarantee robustness toward employee absenteeism, are significant limitations here. These limitations are also compounded by various factors determining workers' productivity, including ergonomic conditions of workplaces (exposure to noise and vibration), levels of perceived fatigue and boredom, as well as risks related to the reduction of qualification levels caused by forgetfulness [8–10]. The last of these factors becomes particularly noticeable in the case of the multi-task dynamic planning of software projects (e.g., scheduling of a portfolio of IT projects) [11], defined as a Dynamic Software Project Scheduling (DSPS) [12,13]. Here, when assigning employees with many qualifications, both the impact of the learning effect



and the forgetting effect on the effectiveness of employee skills must be taken into account. The presented conditions, as well as the need to develop new and/or to extend (improve) existing qualifications, forces the planning of an appropriate job rotation.

The rotation of workplaces requires an appropriate schedule of sequences, matching work requirements and environment to the employees enforced by new orders, while guaranteeing the maintenance of their competences at a fixed (or not lower than assumed) level. In general, this issue also applies to the Hierarchical Worker Assignment Problem (HWAP) [14,15]. It is worth noting that the problem of job rotation understood in this way can be reconstructed as a dynamic matching problem (appearing in many issues occurring at the intersection of economics, operations research and computer science) [16,17] known to be NP-complete.

Since each job requires certain performance-specific skills from the employee assigned to it, in order to reduce the impact of possible employee absenteeism, each assignment of a suitable employee (with appropriate abilities) should be accompanied by care to ensure that they duplicate their duties as a team member. This means that in order to maximize mutual substitutability, when shaping the structure of staff competences, it is necessary to ensure the periodic refreshment of competences of a team's multi-skilled members. The search for an appropriate job rotation is the essence of the Personnel Competence Maintenance Problem (PCMP) [18].

The aforementioned problems overlap and permeate each other. Assuming that the purpose of planning many concurrently implemented projects (undertakings) is such that the management of limited resources (in particular, human resources) guarantees their timely completion, it is easy to notice that various additional restrictions imply different generalizations and/or extensions of the considered dynamic management of concurrently executed order portfolios. The considered problem of the resource-constrained portfolio management observed in this context basically boils down to answering the question of whether there is a non-empty set of solutions (i.e., allocations of employees with the required qualifications) that meet the imposed limitations while following other constraints. An example of such a situation is the adoption of a restriction imposing on the employees' certain qualifications (e.g., nominal level of qualification) and the requirement to maintain their current level of qualifications that are not used (during employment). These restrictions generally result in either the absence of any acceptable solutions or solutions that are not robust toward disruptions related to access to the resources used (e.g., related to cases of employee absenteeism). Of course, there are also further generalizations of the above-presented problem, e.g., extending it to the problem of dynamically scheduling jobs related to the introduction (or exchange) of new resources (e.g., hiring additional employees) or accepting (started in parallel with those still being implemented) new orders.

It seems that an appropriate framework enabling the analytic implementation of that Dynamic Multi-skill Resource-constrained Multi-project Scheduling Problem (DMRPSP), as well as the processes occurring in it of a various nature and character (e.g., learning and forgetting curves and uncertain values of decision variables), will ensure only the adoption of the declarative modeling paradigm. This is because, in the considered class of broadly understood discrete event systems, declarative models allow the representation of nonlinear decision problems (described by logical formulas, algebraic operations, inequalities, etc.) as well as the formulation of decision-making versions of both the problem of analysis (seeking an answer to the question of whether there is a solution that meets the given expectations) and synthesis (seeking an answer to the question under which assumptions there is a solution that meets the given expectations). The presented advantages, combined with the possibility of building models with an open structure implemented in the constraint programming (CP) environment [19], make this representation an attractive alternative to building dedicated DSS class systems.

In the above context, this study refers to our previous work on the assessment of a competence structure's robustness to employee absenteeism and job rotation planning, guaranteeing the maintenance of employees' competences at a given constant level [10,20]. In

turn, this contribution is an attempt to use the previously gained experience for situations in which environmental dynamics play a major role. Examples of this type occur in cases of the dynamic assignment of newly hired employees, or planning new orders while re-scheduling previously established job rotation schedules. Our contribution in this regard includes:

- The DMRPSP declarative reference model integrating DSPS and PCMP problems enabling both the analysis (searching for an answer to the question of whether a required solution is reachable) and synthesis (searching for an answer to the question of which changes to parameters guarantees a required solution).
- Procedures of dynamic software project scheduling aimed at proactive planning with constrained multi-skilled resources, taking into account the influence of learning and forgetting effects.
- The results of many computer experiments illustrating the possibilities of using the presented approach for the rapid prototyping of acceptable job rotation schedules, as well as its scalability.

In summary, the novelty of the proposed approach results from two benefits of adopting the DMRPSP reference model. The first of these results is from its open structure that allows for the further specification and expansion of the model used. The second one boils down to the possibility of formulating and solving synthesis problems related to the search for changes in assumptions (e.g., fields of decision variables), guaranteeing the existence of the expected solution.

The proposed DMRPSP reference model can be implemented in any declarative programming environment (Gurobi, IBM ILOG, Lingo, etc.). The solver built in this way can complement the existing DSS system with project manager support to answer the following questions:

1. What change in the level of employee competences will result in the acceptance of additional orders to the already adopted plan of the project portfolio implementation?
2. Is it possible to accept additional orders during the implementation of the planned portfolio of projects while maintaining the required level of employees' competences?

The organization of the remaining part of the work includes the following. Section 2 discusses selected works in the field of literature and is oriented towards the search for a research gap. Section 3 presents the reference DMRPSP model, and, in particular, its declarative representation and formal (in terms of the problem of the constraint satisfaction) formulation of the problem. The case studies presented in Section 4 illustrate the application of the developed approach to solving problems such as the analysis (regarding the reachability of the expected solution) and synthesis (concerning changes in assumptions guaranteeing the existence of the expected solution). Section 5 summarizes the results of computer experiments conducted to assess the scalability of the proposed approach. Selected directions for future research are collected in Section 6.

## 2. Related Works

Human resources and the related intellectual potential, as well as the skillful sustainable management of them, determine the success of Industry 4.0, and in particular Industry 5.0 [21]. The methods of staff planning, tasks and staff allocation, as well as job rotation, play a key role in this respect. Job rotation plays an especially important role in positions filled by multi-skilled staff [22,23]. Let us note that temporarily delegating an employee to perform tasks consistent with other qualifications or requiring new skills from them enables the gathering of a new experience that extends the existing qualifications [24]. In this context, staffing planning boils down to matching the requirements of the workplace (operation time and competence degree) to the competence and skills in order to maximize work efficiency while maintaining or developing the possessed professional skills and avoiding the boredom of doing the same type of job.

Problems related to employment planning determine the competitiveness of IT companies. In general, the typical software project scheduling problem of assigning the right

employee to the right activity at the right time grows into the problem of scheduling the assignment of multi-skill programmers to a portfolio of concurrently implemented projects [25,26]. Assuming the multi-competence of team members, it is natural to distinguish different levels of competences. Assuming, in turn, that a programmer with a higher level of competence can replace another team member with a lower level of this competence, it is possible to plan substitutions, which is already the subject of the hierarchical worker assignment problem [14,15].

These problems, in turn, naturally extend to the dynamic multi-skill resource-constrained multi-project scheduling problem, in which new orders are taken up during the execution of orders started earlier, as soon as the resources necessary for their implementation become available. Its other extensions, e.g., taking into account the effects of learning and forgetting, make it possible to formulate questions regarding the conditions guaranteeing the maintenance of the level of competence of the employed team [27]. All the problems of staffing, personnel allocation and job assignment, as well as the job rotation scheduling presented above, are instances of a more general problem, which is the dynamic matching problem [28].

Many works have been devoted to the staff assignment problem taking into account learning and forgetting effects [29,30]. Learning and forgetting curves illustrate the correlations between learning outcomes (e.g., the level of skills acquired) and the frequency of repetition to achieve them. Thus, according to the learning curve theory, the more often an individual repeats a process or activity, the more adept they become at that activity [31]. In turn, according to the forgetting curve theory, individuals quickly lose the knowledge they have acquired. This means that in order to preserve the acquired knowledge, they must refresh it by actively reviewing the previously learned material. Consequently, cultivating acquired skills, especially in multi-skilled teams, requires their systematic verification and refreshment. The relevant capability maintenance schedule can be established on the basis of appropriate learning and forgetting curve formulas. Of course, in order to personify them appropriately, this process should be preceded by an experimental identification of the functions' describing curves. Thus, it is worth noting that activities focused on cultivating skills (i.e., maintaining a given level of competence) can find their practical expression in job rotation schedules, used by multi-skilled teams during the implementation of a project's portfolio.

Staffing multi-skilled workers increases the company's robustness (e.g., to employees' absenteeism), but makes it difficult to perform the related project portfolio scheduling and staff assignment. These difficulties increase even more when assigning multi-skilled workers, taking into account the effect of learning and forgetting on their effectiveness. Problems of this type usually occur during the scheduling of an IT project's portfolio. These problems, emphasizing various issues such as software multi-project resource scheduling [32], activity scheduling in the dynamic, multi-project setting [33–35], human resource allocation in a multi-project [36] or Resource Constrained Project Scheduling Problem (RCPSP) [37–39] are part of the dynamic resource constrained multi-project management; in other words, in DMRPSP. Consequently, since RCPSP is NP-hard, so is DMRPSP.

The diversity of models and methods used to solve the above-mentioned problems results from their nature and related specificity. They include stochastic models (aimed at, e.g., risk assessment) [40], operational research models (e.g., based on integer or mixed-integer programming, dynamic programming, etc.) [41], simulation models (e.g., based on the multi-agent concept) [42] and artificial intelligence models and fuzzy models (using, e.g., population algorithms, metaheuristics, fuzzy logic algorithms, etc.) [43,44]. Formal representations of these models implemented in the relevant methods of imperative programming are useful in solving the so-called "situation analysis problem", i.e., related to the search for an answer to the question of whether a given set of decision variables guarantee a specific (extreme) value of the objective function. This limitation is not introduced by the declarative programming paradigm. In addition to the possibility of formulating and solving issues such as the "situation analysis problem", it also allows for the "situation synthesis problem". The problem of the so-called situation synthesis, in

turn, allows for the search for decision variables domains at which a given value of the objective function holds.

Models implementing the declarative programming paradigm have the greatest chance of meeting these expectations. The constraints' programming strategies used in them enable both the formulation of analysis and synthesis problems, which result from their nature, i.e., because these models inherently have open structures.

Unfortunately, the declarative approach is very rarely used either for modeling or solving DMRPSP-like problems. This deficiency is visible in, among others, the lack of studies resulting in the development of conditions, the fulfillment of which guarantees the existence of acceptable solutions while covering problems of dynamic staffing and tasks allocation, job rotation scheduling guaranteeing the maintenance of staff competences at assumed level, dynamic multi-project scheduling, etc. The proposed declarative model makes it possible to determine whether any solution to the admissible problem formulated in it exists. In the absence of solutions (meeting the given constraints), it enables for determining the change of the relevant constraints to such that the set of admissible solutions is not empty. The presented research gap is confirmed by the relatively few studies [45]. It is easy to observe that filling this gap will contribute to the creation of systems supporting the software project manager in the dynamic planning of resource-constrained multi-projects in the IT environment.

## 3. Model Formulation

### 3.1. Illustrative Example

In order to introduce the issue of portfolio scheduling, taking into account the loss of competences of project team members, let us consider the following illustrative example shown in Figure 1. The project portfolio $E$ consists of two projects, $E_1$ and $E_2$, with a known precedence between them. Each project contains several tasks. The $E_1$ project requires the implementation of tasks $Z_1$, $Z_2$ and $Z_4$, while the project $E_2$ requires tasks $Z_1$, $Z_2$, $Z_3$ and $Z_4$. The completion time of each project is limited by the $HE_i$ time horizon allocated to it. In the case under consideration, $HE_1 = 4$ for the project $E_1$, and $HE_2 = 5$ for the project $E_2$.

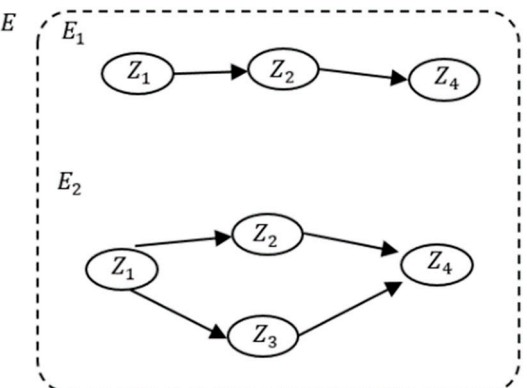

**Figure 1.** Project portfolio activity network $E$.

The available team consists of three employees $P = \{P_1, P_2, P_3\}$. The competences of the team members are listed in the $G^0$ competence structure; see Table 1.

**Table 1.** $G^0$ competence structure of team $P$ employees performing tasks $Z$.

| $G^0$ | $P_1$ | $P_2$ | $P_3$ |
|---|---|---|---|
| $Z_1$ | 3 | 5 | 3 |
| $Z_2$ | 4 | 4 | 4 |
| $Z_3$ | 5 | 4 | 4 |
| $Z_4$ | 4 | 4 | 5 |

In general, the competence structure is defined as the matrix $G^n = \left[g_{k,i}^n\right]_{k=1\ldots M;i=1\ldots Q}$, whose elements $g_{k,i}^n$ determine the level of competence of the $k$-th employee for the $i$-th task, in the $n$-th moment of time. In the example under consideration, $g_{k,i}^n \in \{1,\ldots,5\}$. In the presented case, Table 1 means that before starting the implementation of projects (in the 0-th unit of time), the employee doing $P_1$ has the competence to $Z_1$ task at level 3 ($g_{1,1}^0 = 3$), to task $Z_2$ at level 4 ($g_{1,2}^0 = 4$), to task $Z_3$ at level 5 ($g_{1,3}^0 = 5$), etc. The adopted values of $g_{k,i}^n$ determine the time of task execution according to the following principle:

$$g_{k,i}^n = \begin{cases} 5 \text{ or } 4 & \text{employee } P_k \text{ performs the task } Z_i \text{ in } t_i = 1 \text{ time unit} \\ 3 \text{ or } 2 \text{ or } 1 & \text{employee } P_k \text{ performs the task } Z_i \text{ in } t_i = 2 \text{ time units} \end{cases}.$$

According to this principle, for the competence structure from Table 1, the times of task implementation are given in Table 2. It shows that, for example, an employee $P_1$ performs a task $Z_1$ during 2 units, and tasks $Z_2$, $Z_3$, $Z_4$ during 1 unit, etc.

**Table 2.** Execution times of tasks of the $Z$ set by employees of the $P$ team.

|  | $P_1$ | $P_2$ | $P_3$ |
|---|---|---|---|
| $Z_1$ | 2 | 1 | 2 |
| $Z_2$ | 1 | 1 | 1 |
| $Z_3$ | 1 | 1 | 1 |
| $Z_4$ | 1 | 1 | 1 |

The assignment of workers to tasks is represented in the form of a matrix $X^n = \left[x_{k,i,j}^n\right]_{k=1\ldots K;i=1\ldots Q;j=1\ldots J}$, where: $x_{k,i}^n \in \{0,1\}$, $x_{k,i}^n = 1$ when the employee $P_k$ performs the task $Z_i$ of the $E_j$ project in $n$-th moment time; otherwise, $x_{k,i}^n = 0$. The following rules for assigning employees to the following tasks shall be adopted:

- Only one employee can be assigned to one task at a time.
- One employee can perform different tasks in the portfolio.
- An employee can perform only one task per unit of time.

An example of the assignment of team $P$ employees to the tasks of set $Z$ is presented in Table 3. This means that in the first moment of time, the employee $P_1$ performs the task $Z_1$, the employee $P_2$ performs the task $Z_4$ and the employee $P_3$ performs the task $Z_2$.

**Table 3.** Assignment of $X^1$ employees of team $P$ to perform the tasks of the $Z$ set.

| $X^1$ | $P_1$ | $P_2$ | $P_3$ |
|---|---|---|---|
| $Z_1$ | 1 | 0 | 0 |
| $Z_2$ | 0 | 0 | 1 |
| $Z_3$ | 0 | 0 | 0 |
| $Z_4$ | 0 | 1 | 0 |

According to the learning and forgetting effect, employees who do not provide a specific task lose the knowledge and skills related to it after some period of time. Curves that model this effect are the subject of many studies [9,27], in which different shapes are given with different degrees of inclination. For simplicity, we assume that the competence level:

- Increases by 1 after the task is completed by a specific employee.
- Decreases by 1 for every 2 units of time when an employee does not perform a task requiring this competence.

In other words, the level of employees' competences changes during the implementation of subsequent projects. This means that the allocation $X$ adopted to perform the tasks of portfolio $E$ affects the structure of competences of the employees involved in its

implementation. To this end, the concept of competence structure degree $SG^n$ is introduced, which is the sum of competence levels $g_{k,i}^n$ of the $G^n$ structure. Consequently, for the $G^0$ structure in Table 1, the competence structure degree follows $SG^0 = 49$.

Under the influence of the learning effect and forgetting, the degree of competence structure $SG^n$ changes over time, i.e., this degree can:

- Decrease to the minimum degree $MINSG = K \times Q \times g_{MIN} = 3 \times 4 \times 1 = 12$, where $K$—number of employees, $Q$—number of tasks and $g_{MIN}$—minimum level of competence.
- Increase to the maximum degree $MAXSG = K \times Q \times g_{MAX} = 3 \times 4 \times 5 = 60$, where $K$—number of employees, $Q$—number of tasks and $g_{MAX}$—maximum level of competence.

In the present case, $MINSG = 12$ and $MAXSG = 60$. It is arbitrarily assumed that after the completion of the project portfolio, the degree of competence structure should be $SG \geq 25$ (i.e., around 25% of the maximum value).

Assuming these assumptions, an answer to the following question is sought: Is there such a sequence of assignments $X^1$, $X^2$, ... , $X^n$ that guarantees the execution of the portfolio of projects $E$ in the given horizons $HE_1 = 4$ and $HE_2 = 5$ while maintaining the degree of competence structure $SG \geq 25$?

Figure 2 illustrates two examples of project delivery variants, which differ in the method of task assignment, without rotation and with rotation:

- In variant (a), the rank $SG^5 = 32$ was obtained, which corresponds to the situation where employees perform tasks without rotation, i.e., a specific task is carried out by only one employee; for example, a task $Z_1$ performs only $P_2$, a task $Z_2$ performs only $P_1$, etc.
- In variant (b), the rank $SG^5 = 34$ is obtained, which corresponds to the situation where employees perform tasks with rotation, i.e., a specific task is carried out by different employees; for example, a task $Z_1$ performs $P_1$ and $P_2$, a task $Z_2$ performs $P_2$, and $P_3$, etc.

It follows that without rotation, a lower level of competence structure has been achieved, but this does not threaten the timely implementation of the project portfolio.

The degree of competence structure $SG^n$ proves to be crucial when unplanned events occur during the implementation of projects, hereinafter referred to as disruptions. For example, suppose that the disruption under consideration takes the form of an additional $E_3$ project (network of activities as shown in Figure 3), which is expected to start when projects $E_1$ and $E_2$ are completed.

An answer to the question is sought: For both early variants' (a) and (b) implementation of the project portfolio, is it possible to accept an additional project order $E_3$, while maintaining the level of competence structure $SG^n \geq 25$?

As can be observed in Figure 4:

- In variant (a), the project $E_3$ ends in the eighth unit of time, and the degree $SG^8 = 22$.
- In variant (b), the project $E_3$ ends in the seventh unit of time, and the degree $SG^7 = 27$.

This means that the rotation approach enables the implementation of a portfolio of projects and the maintenance of staff competences in the case of new orders. It should be noted that a higher level of competence structure also translates into a reduction in the time of task implementation (compare the shorter time to complete projects in variant (b)).

Variant (a) without task rotation:

project $E_j$    task $Z_i$    employee $P_k$    project horizon $HE_2$    project horizon $HE_1$

assignment $X^n$    competence structure $G^n$

Gantt chart (Variant a):

- $E_2$: $Z_4 (P_3)$; $Z_3 (P_2)$; $Z_2 (P_1)$; $Z_1 (P_2)$
- $E_1$: $Z_4 (P_3)$; $Z_2 (P_1)$; $Z_1 (P_2)$
- t.u. axis: 0 1 2 3 4 5

**Assignments $X^n$ (Variant a):**

| $X^1$ | $P_1$ | $P_2$ | $P_3$ |
|---|---|---|---|
| $Z_1$ | 0 | 1 | 0 |
| $Z_2$ | 0 | 0 | 0 |
| $Z_3$ | 0 | 0 | 0 |
| $Z_4$ | 0 | 0 | 0 |

| $X^2$ | $P_1$ | $P_2$ | $P_3$ |
|---|---|---|---|
| $Z_1$ | 0 | 1 | 0 |
| $Z_2$ | 1 | 0 | 0 |
| $Z_3$ | 0 | 0 | 0 |
| $Z_4$ | 0 | 0 | 0 |

| $X^3$ | $P_1$ | $P_2$ | $P_3$ |
|---|---|---|---|
| $Z_1$ | 0 | 0 | 0 |
| $Z_2$ | 1 | 0 | 0 |
| $Z_3$ | 0 | 1 | 0 |
| $Z_4$ | 0 | 0 | 1 |

| $X^4$ | $P_1$ | $P_2$ | $P_3$ |
|---|---|---|---|
| $Z_1$ | 0 | 0 | 0 |
| $Z_2$ | 0 | 0 | 0 |
| $Z_3$ | 0 | 1 | 0 |
| $Z_4$ | 0 | 0 | 0 |

| $X^5$ | $P_1$ | $P_2$ | $P_3$ |
|---|---|---|---|
| $Z_1$ | 0 | 0 | 0 |
| $Z_2$ | 0 | 0 | 0 |
| $Z_3$ | 0 | 0 | 0 |
| $Z_4$ | 0 | 0 | 1 |

**Competence structures $G^n$ (Variant a):**

| $G^0$ | $P_1$ | $P_2$ | $P_3$ |
|---|---|---|---|
| $Z_1$ | 3 | 5 | 3 |
| $Z_2$ | 4 | 4 | 4 |
| $Z_3$ | 5 | 4 | 4 |
| $Z_4$ | 4 | 4 | 5 |

| $G^1$ | $P_1$ | $P_2$ | $P_3$ |
|---|---|---|---|
| $Z_1$ | 3 | 5 | 3 |
| $Z_2$ | 4 | 4 | 4 |
| $Z_3$ | 5 | 4 | 4 |
| $Z_4$ | 4 | 4 | 5 |

| $G^2$ | $P_1$ | $P_2$ | $P_3$ |
|---|---|---|---|
| $Z_1$ | 2 | 5 | 2 |
| $Z_2$ | 5 | 3 | 3 |
| $Z_3$ | 4 | 3 | 3 |
| $Z_4$ | 3 | 3 | 4 |

| $G^3$ | $P_1$ | $P_2$ | $P_3$ |
|---|---|---|---|
| $Z_1$ | 2 | 5 | 2 |
| $Z_2$ | 5 | 3 | 3 |
| $Z_3$ | 4 | 3 | 3 |
| $Z_4$ | 3 | 3 | 5 |

| $G^4$ | $P_1$ | $P_2$ | $P_3$ |
|---|---|---|---|
| $Z_1$ | 1 | 4 | 1 |
| $Z_2$ | 5 | 2 | 2 |
| $Z_3$ | 3 | 4 | 2 |
| $Z_4$ | 2 | 2 | 5 |

| $G^5$ | $P_1$ | $P_2$ | $P_3$ |
|---|---|---|---|
| $Z_1$ | 1 | 4 | 1 |
| $Z_2$ | 4 | 2 | 2 |
| $Z_3$ | 3 | 4 | 2 |
| $Z_4$ | 2 | 2 | 5 |

Variant (b) with task rotation:

project $E_j$    task $Z_i$    employee $P_k$    project horizon $HE_2$    project horizon $HE_1$

assignment $X^n$    competence structure $G^n$

Gantt chart (Variant b):

- $E_2$: $Z_4 (P_3)$; $Z_3 (P_1)$; $Z_2 (P_2)$; $Z_1 (P_1)$
- $E_1$: $Z_4 (P_3)$; $Z_2 (P_3)$; $Z_1 (P_2)$
- t.u. axis: 0 1 2 3 4 5

**Assignments $X^n$ (Variant b):**

| $X^1$ | $P_1$ | $P_2$ | $P_3$ |
|---|---|---|---|
| $Z_1$ | 1 | 1 | 0 |
| $Z_2$ | 0 | 0 | 0 |
| $Z_3$ | 0 | 0 | 0 |
| $Z_4$ | 0 | 0 | 0 |

| $X^2$ | $P_1$ | $P_2$ | $P_3$ |
|---|---|---|---|
| $Z_1$ | 1 | 0 | 0 |
| $Z_2$ | 0 | 0 | 1 |
| $Z_3$ | 0 | 0 | 0 |
| $Z_4$ | 0 | 0 | 0 |

| $X^3$ | $P_1$ | $P_2$ | $P_3$ |
|---|---|---|---|
| $Z_1$ | 0 | 0 | 0 |
| $Z_2$ | 0 | 1 | 0 |
| $Z_3$ | 1 | 0 | 0 |
| $Z_4$ | 0 | 0 | 1 |

| $X^4$ | $P_1$ | $P_2$ | $P_3$ |
|---|---|---|---|
| $Z_1$ | 0 | 0 | 0 |
| $Z_2$ | 0 | 1 | 0 |
| $Z_3$ | 0 | 0 | 0 |
| $Z_4$ | 0 | 0 | 0 |

| $X^5$ | $P_1$ | $P_2$ | $P_3$ |
|---|---|---|---|
| $Z_1$ | 0 | 0 | 0 |
| $Z_2$ | 0 | 0 | 0 |
| $Z_3$ | 0 | 0 | 0 |
| $Z_4$ | 0 | 0 | 1 |

**Competence structures $G^n$ (Variant b):**

| $G^0$ | $P_1$ | $P_2$ | $P_3$ |
|---|---|---|---|
| $Z_1$ | 3 | 5 | 3 |
| $Z_2$ | 4 | 4 | 4 |
| $Z_3$ | 5 | 4 | 4 |
| $Z_4$ | 4 | 4 | 5 |

| $G^1$ | $P_1$ | $P_2$ | $P_3$ |
|---|---|---|---|
| $Z_1$ | 3 | 5 | 3 |
| $Z_2$ | 4 | 4 | 4 |
| $Z_3$ | 5 | 4 | 4 |
| $Z_4$ | 4 | 4 | 5 |

| $G^2$ | $P_1$ | $P_2$ | $P_3$ |
|---|---|---|---|
| $Z_1$ | 4 | 5 | 2 |
| $Z_2$ | 3 | 3 | 5 |
| $Z_3$ | 4 | 3 | 3 |
| $Z_4$ | 3 | 3 | 4 |

| $G^3$ | $P_1$ | $P_2$ | $P_3$ |
|---|---|---|---|
| $Z_1$ | 4 | 4 | 2 |
| $Z_2$ | 3 | 3 | 5 |
| $Z_3$ | 5 | 3 | 3 |
| $Z_4$ | 3 | 3 | 5 |

| $G^4$ | $P_1$ | $P_2$ | $P_3$ |
|---|---|---|---|
| $Z_1$ | 3 | 4 | 1 |
| $Z_2$ | 2 | 4 | 4 |
| $Z_3$ | 5 | 2 | 2 |
| $Z_4$ | 2 | 2 | 5 |

| $G^5$ | $P_1$ | $P_2$ | $P_3$ |
|---|---|---|---|
| $Z_1$ | 3 | 3 | 1 |
| $Z_2$ | 2 | 4 | 4 |
| $Z_3$ | 4 | 2 | 2 |
| $Z_4$ | 2 | 2 | 5 |

**Figure 2.** Variants for the implementation of the project portfolio without task rotation (**a**) and with task rotation (**b**).

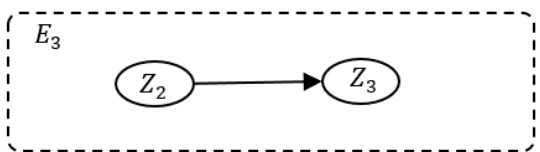

**Figure 3.** The network of project activities $E_3$.

Variant (a) without task rotation:

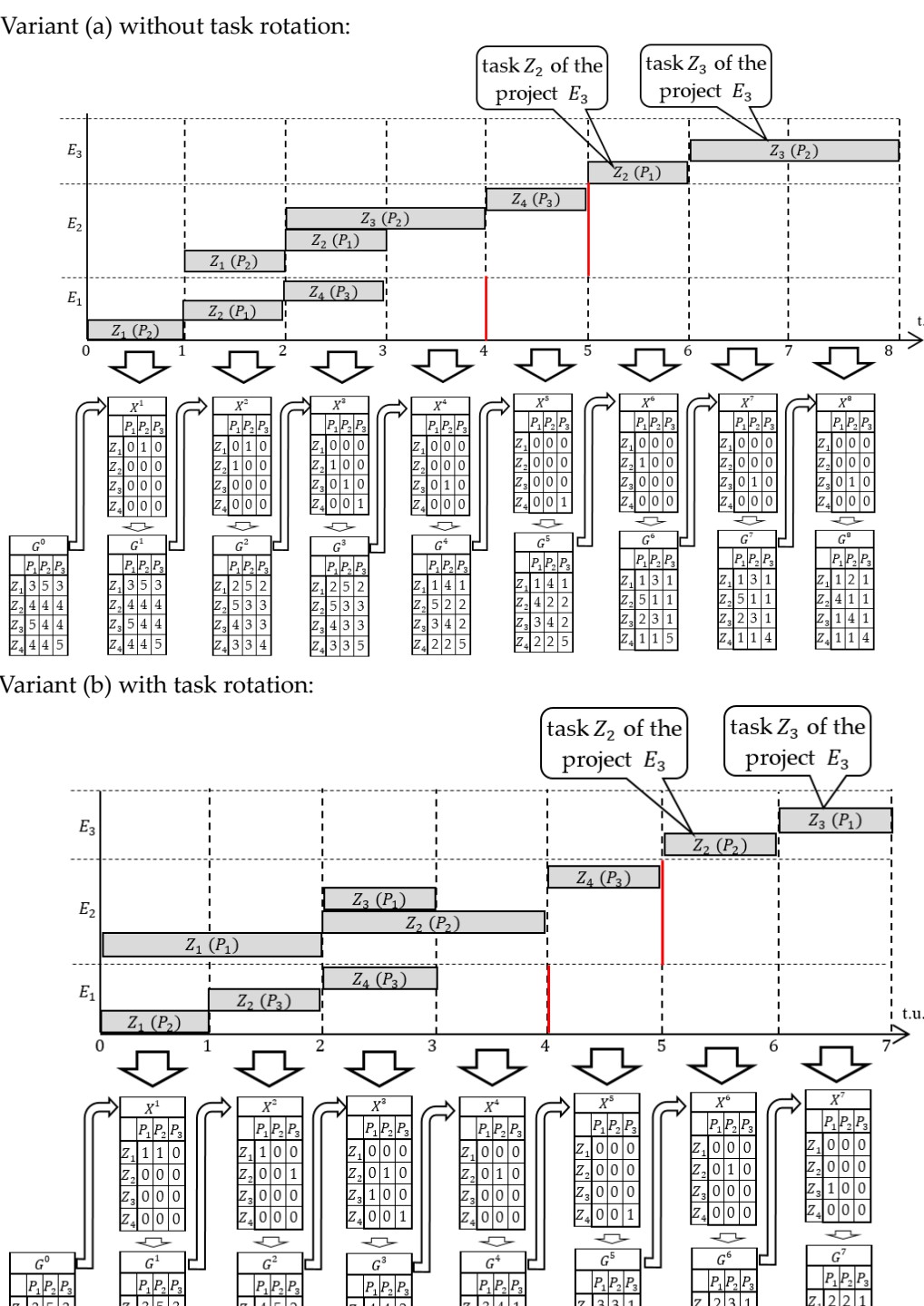

**Figure 4.** Variants for the implementation of the project portfolio without task rotation (**a**), and with task rotation (**b**), with a new project commissioned $E_3$.

### 3.2. Declarative Model Description

The considered problem could be described using the declarative modeling paradigm.

**Parameters**:

$Z$ : set of tasks, $Z = \{Z_1, \ldots, Z_i, \ldots, Z_Q\}$;
$EP$ : set of planned projects;

$EX$ : set of additional orders (disruptions);

$E$ : portfolio of projects, $E = EP \cup EX = \{E_1, \ldots, E_j, \ldots, E_J\}$, $|E| = J$;

$E_j$: $j$-th project of portfolio $E$, represented by the network of activities: $E_j = (VE_j, AE_j)$, where $VE_j \subseteq Z$ is a set of tasks of the project $E_j$ and $AE_j$ is a set of arcs defining order relations $AE_j \subseteq VE_j \times VE_j$;

$HE_j$: latest project completion date $E_j$;

$SE_j$: earliest project start date $E_j$;

P : team of programmers, $P = \{P_1, \ldots, P_k, \ldots, P_K\}$;

$G^0$: the initial structure of competences;

$SG^0$: the initial degree of the competence structure;

$\varphi$: function determining the level of competence of programmers $\varphi(G^n, X^n)$;

$\tau_{j,i}$: function determining the time of task $Z_i$ from project $E_j$ depending on the competence of the employee assigned to it, $\tau_{j,i}(G^n, X^n)$;

$SG^*$: the expected degree of competence structure.

**Decision variables:**

$Y_j$: schedule for $j$-th project execution, $Y_j = (y_{j,1}, \ldots, y_{j,i}, \ldots, y_{j,Q})$, where $y_{j,i}$—the start of the task $Z_i$ from the $E_j$ project;

$X^n$: assignment of team $P$ employees to tasks $Z$ at moment $n$ : $X^n = \left[ x_{k,i,j}^n \right]_{k=1\ldots K; i=1\ldots Q; j=1\ldots J}$;

$G^n$: competence structure at moment $n$: $G^n = \left[ g_{k,i}^n \right]_{k=1\ldots K; i=1\ldots Q}$.

**Constraints:**

- The tasks must be performed according to the sequence specified by the $E_j$, project activity network, and within the time limit specified by $SE_j$ and $HE_j$:

$$y_{j,\alpha} = 0, \text{ if } Z_\alpha \notin VE_j, \ \forall E_j \in E \tag{1}$$

$$y_{j,\alpha} = y_{j,\beta} + \tau_{j,\beta}(G^n, X^n), \text{ if } (Z_\alpha, Z_\beta) \in AE_j, \ \forall E_j \in E \tag{2}$$

$$y_{j,\alpha} \geq SE_j, \text{ if } Z_\alpha \in VE_j, \ \forall E_j \in E \tag{3}$$

$$y_{j,\alpha} + \tau_{j,\alpha}(G^n, X^n) \leq HE_j, \text{ if } Z_\alpha \in VE_j, \ \forall E_j \in E \tag{4}$$

- The assignment of a worker $P_k$ to a $Z_i$ task determines when it starts:

$$\left( x_{k,i,j}^n = 1 \right) \Rightarrow \left( y_{j,i} = n \right), \ \forall P_k \in P, \ \forall Z_i \in Z, \ \forall E_j \in E, \ n = 0 \ldots \max_{j=1\ldots J}\{HE_j\} \tag{5}$$

$$\left( y_{j,i} = n \right) \Rightarrow \left( \sum_{k=1}^{K} x_{k,i,j}^n = 1 \right), \ \forall Z_i \in Z, \ \forall E_j \in E, \ n = 0 \ldots \max_{j=1\ldots J}\{HE_j\} \tag{6}$$

- At a time $n$, each employee $P_k$ is assigned to only one task:

$$\sum_{i=1}^{Q} \sum_{j=1}^{J} x_{k,i,j}^n \leq 1, \ \forall P_k \in P, \ n = 0 \ldots \max_{j=1\ldots J}\{HE_j\} \tag{7}$$

- Each task $Z_i$ and the project $E_j$ must be completed:

$$\sum_{n=0}^{HE_j} \sum_{k=1}^{K} x_{k,i,j}^n = 1, \text{ if } \forall Z_i \in VE_j, \ \forall E_j \in E \tag{8}$$

- The structure of competences changes over time (the effect of forgetting and learning):

$$g_{k,i}^n = g_{k,i}^{n-1} + \varphi(G^n, X^n), \ \forall P_k \in P, \ \forall Z_i \in Z, \ n = 1 \ldots \max_{j=1\ldots J}\{HE_j\} \tag{9}$$

- The level of the competence structure after the completion of the project portfolio cannot be less than the set value $SG^*$:

$$SG^n = \sum_{k=1}^{K} \sum_{i=1}^{Q} g_{k,i}^n \tag{10}$$

$$SG^{\max_{j=1\ldots J}\{HE_j\}} \geq SG^* \tag{11}$$

Among the introduced constraints can be distinguished restrictions (1)–(8) describing the relationship between the assignment of employees to tasks and their schedule, as well as restrictions (9)–(11) describing the impact of task assignment on the structure of competences and the impact of the competence structure on the duration of tasks.

*3.3. Problem Statement*

The presented model allows for answering the following questions:

- What change in the level of competence structure of $SG^n$ will result in the acceptance of additional $EX$ orders to the already adopted $EP$ collection project plan? This occurs in the so-called analysis problem.
- Is it possible to accept additional $EX$ orders for a successful $EP$ portfolio while maintaining the required level of competence structure $SG^*$? This occurs in the so-called synthesis problem.

Related problems can be formulated in terms of the following Constraint Satisfaction Problem (CSP):

$$CS = ((\mathcal{V}, \mathcal{D}), \mathcal{C}) \tag{12}$$

where: $\mathcal{V} = \left\{ y_{j,i}, x_{k,i,j}^n, g_{k,i}^n \,\middle|\, k = 1, \ldots, K; \ i = 1, \ldots, Q; \ j = 1, \ldots, J \right\}$, a set of decision variables representing schedule $Y_j$, assignment $X^n$ and competence structure $G^n$; $\mathcal{D}$ is a finite set of domains of decision variables; $\mathcal{C}$ is a set of constraints specified in inequalities (1)–(11).

To solve problem $CS$ (12), one should determine such values of decision variables $y_{j,i}, x_{k,i,j}^n, g_{k,i}^n$, for which all the constraints given in the set $\mathcal{C}$ are satisfied. Solving $CS$ means determining the schedule $Y_j$, assignment $X^n$, and competence structure $G^n$ which guarantees the project portfolio $EP$ execution with a given level of $SG^*$ for occurrence in new projects $EX$.

The proposed $CS$ model (12) is part of an iterative (alternating) method of solving analysis and/or synthesis problems [20]. The essence of this method is a scheme of alternating approaches. To solve analysis problems, use the question:

- Is there an admissible solution for the input data (competence structure with a given competence level)?

    For synthesis problems, use the question:

- Are there other inputs (e.g., assignment of workers to tasks) resulting in an admissible solution (i.e., required level of competence structure)?

    This method was used in the experiments described in the next section.

## 4. Case Study

This case study describes the project portfolio implementation in the IT company carrying out orders covering the development of mobile applications. A portfolio of two projects is given: $EP = \{E_1, E_2\}$, whose network activities are presented in Figure 5. Projects must be completed by $HE_1 = 9$, $HE_2 = 10$. To perform the tasks $Z = \{Z_1, \ldots, Z_5\}$,

three employees were employed $P = \{P_1, P_2, P_3\}$ with competences adopted in the $G^0$ competence structure from Table 4.

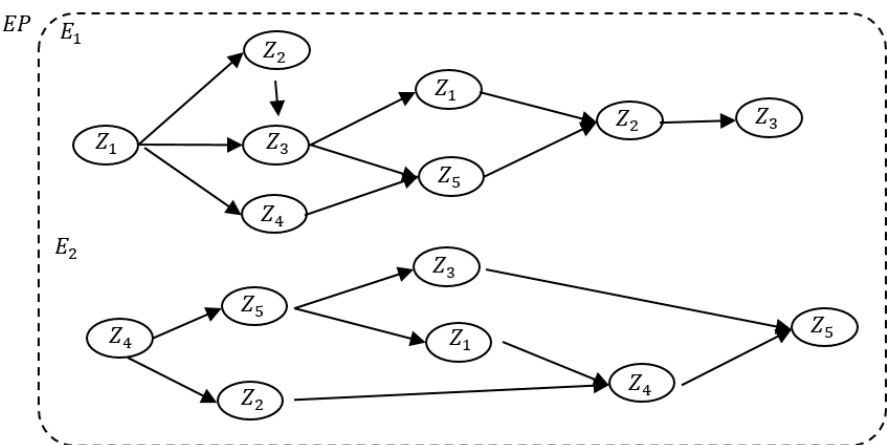

**Figure 5.** Project portfolio activity network $EP$.

**Table 4.** Initial structure of the competence structure $G^0$.

| $G^0$ | $P_1$ | $P_2$ | $P_3$ |
|---|---|---|---|
| $Z_1$ | 5 | 2 | 1 |
| $Z_2$ | 4 | 4 | 4 |
| $Z_3$ | 5 | 4 | 3 |
| $Z_4$ | 2 | 4 | 5 |
| $Z_5$ | 1 | 3 | 1 |

It was assumed that the time of execution of tasks belonging to the $Z$ set depends on the level of competence of employees and is set by the following function:

$$\tau_{j,i} = \begin{cases} \text{the task lasts 1 time unit if the competence level is 5} \\ \text{the task lasts 2 time unit if the competence level is 4} \\ \text{the task lasts 3 time unit if the competence level is 3} \\ \text{the task lasts 4 time unit if the competence level is 2} \\ \text{the task lasts 5 time unit if the competence level is 1} \end{cases}$$

The level of an employee's competences, in turn, changes over time depending on the state of their current tasks, i.e., according to the following function:

$$\varphi = \begin{cases} \text{level of competence decreases by 1 every 5 units of time} \\ \text{level of competence increases by 1 when the employee starts the task} \end{cases}$$

The assumed initial degree of the competence structure is $SG^0 = 48$.

It was also assumed that the $EP$ portfolio under consideration follows the following sequence of assignments' $X^1$, $X^2$, ..., $X^{10}$ (i.e., assignments of team $P$ employees to tasks $Z$ in consecutive units of time) allocations, which guarantees the timely execution of the projects comprising this portfolio (within the accepted deadlines: $HE_1 = 9$, $HE_2 = 10$)—see Figure 6. Task assignments are carried out without rotation, i.e., task $Z_1$ is performed only by an employee $P_1$, task $Z_2$ only by $P_3$, etc. At the end of the $EP$ project portfolio, the degree of competence structure is $SG^{10} = 35$ (Table 5).

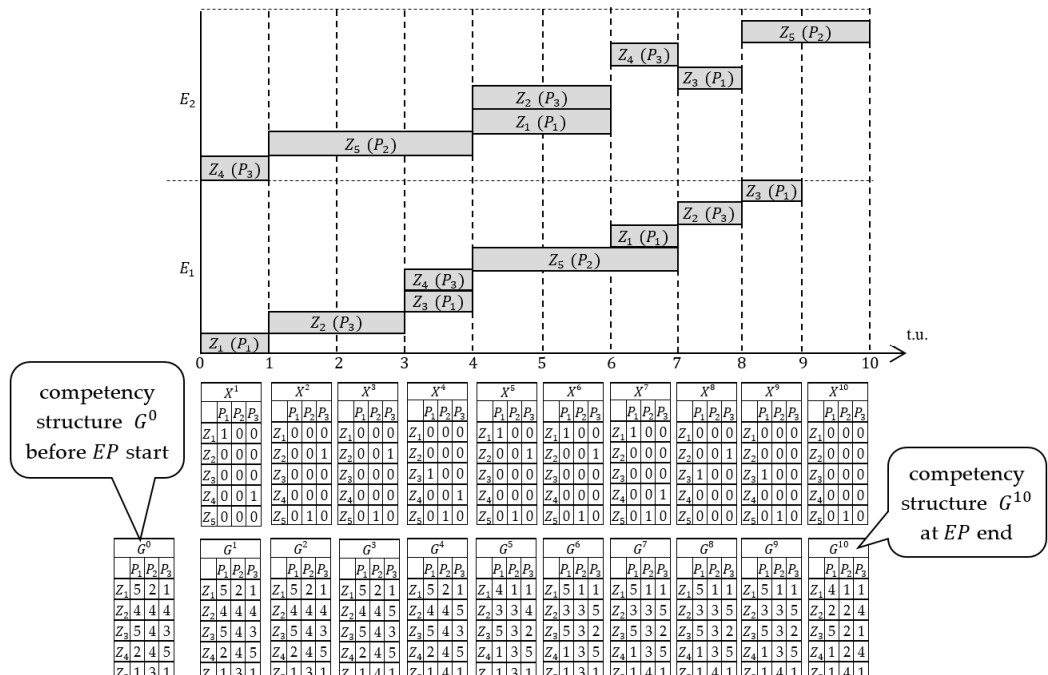

**Figure 6.** Sequence of assignments $X^1, X^2, \ldots, X^{10}$ used to implement the *EP* project portfolio.

**Table 5.** Competence structure $G^{10}$ after completion of the *EP* project portfolio, with the degree $SG^{10} = 35$.

| $G^{10}$ | $P_1$ | $P_2$ | $P_3$ |
|---|---|---|---|
| $Z_1$ | 4 | 1 | 1 |
| $Z_2$ | 2 | 2 | 4 |
| $Z_3$ | 5 | 2 | 1 |
| $Z_4$ | 1 | 2 | 4 |
| $Z_5$ | 1 | 4 | 1 |

Only those projects (implemented in the form of additional orders) are accepted for the *EP* portfolio, the implementation of which will not cause the level of the competence structure to fall below 35: $SG^* = 35$ .

### 4.1. Analysis of Possible Solutions

A new *EX* order consisting of one project $E_3$ ($EX = \{E_3\}$) has been introduced to the *EP* portfolio, the structure of which is illustrated in Figure 7. The project should be completed within the specified time interval: $[SE_3, HE_3] = [5, 17]$. For such a portfolio, $E = EP \cup EX$, the answer to the following question (analysis problem) is sought: What change in the level of the competence structure of $SG^{10}$ will result in the acceptance of additional *EX* orders to the already adopted *EP* collection project plan?

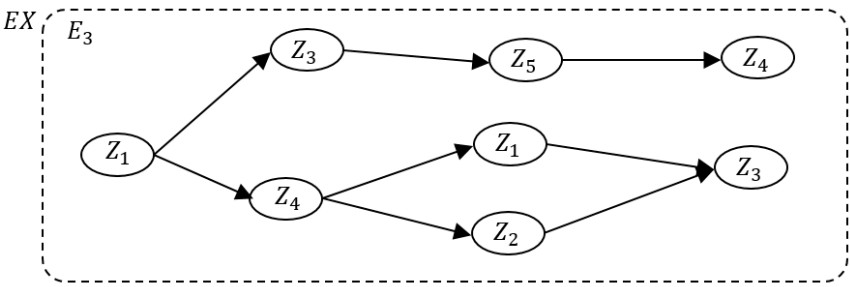

**Figure 7.** Project activity network $E_3$.

The problem under consideration was formulated as *CS* (12) and solved in the IBM ILOG environment. An example of an acceptable solution is shown in Figure 8. It confirms the ability to complete the *EP* project portfolio and the additional *EX* order within $HE_3 = 17$. However, this reduces the competence structure level to $SG^{17} = 33$, which is lower than assumed: $SG^{17} \geq SG^* = 35$.

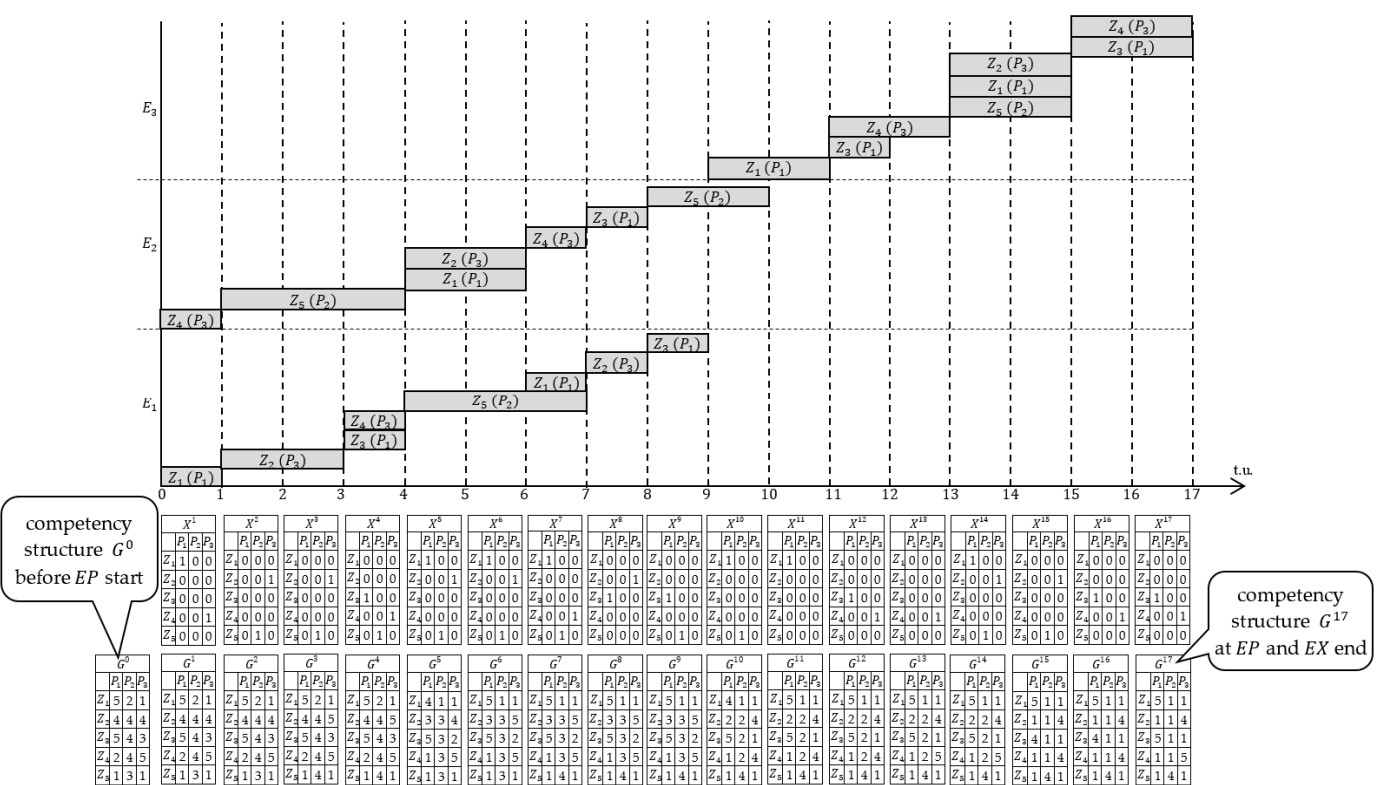

**Figure 8.** Solution to the analysis problem in which the portfolio *EP* was executed according to the sequence of assignments $X^1, X^2, \ldots, X^{17}$ without task rotation but, however, with an additional *EX* order.

The resulting solution is characterized by assignments without task rotation. Therefore, there is a situation where employees have specialized in certain activities at the expense of losing competence in others. It should be noted that ordering a new project during the implementation of previously planned tasks (*EP* portfolio projects) gives the opportunity to already introduce task rotation schedules at the stage of their implementation.

### 4.2. Synthesis of Expected Solutions

The search for task rotation schedules boils down to the answer to the following question (see the synthesis problem formulation): Is there a task rotation schedule that guarantees both the *EP* project portfolio and the execution of additional *EX* orders while maintaining the required degree level of competence structure $SG^* \geq 35$?

As in the previous case, the problem under consideration was formulated as *CS* (12) and solved in the IBM ILOG environment. An example of an acceptable solution is shown in Figure 9. The considered EP project portfolio is executed on time, as well as the additionally accepted *EX* order within $HE_3 = 17$ while maintaining the expected level of competence structure: $SG^{17} = 36$. The result obtained is a consequence of the task rotation used. For example, already at the first moment ($t = 5$) of the project implementation, the assignment $X^5$ changed in relation to the assignment from Figure 8—the employee was $P_3$ assigned to the new task $Z_3$ (previously $Z_2$). This type of rotation (carried out cyclically) allows for maintaining the required level of competence structure.

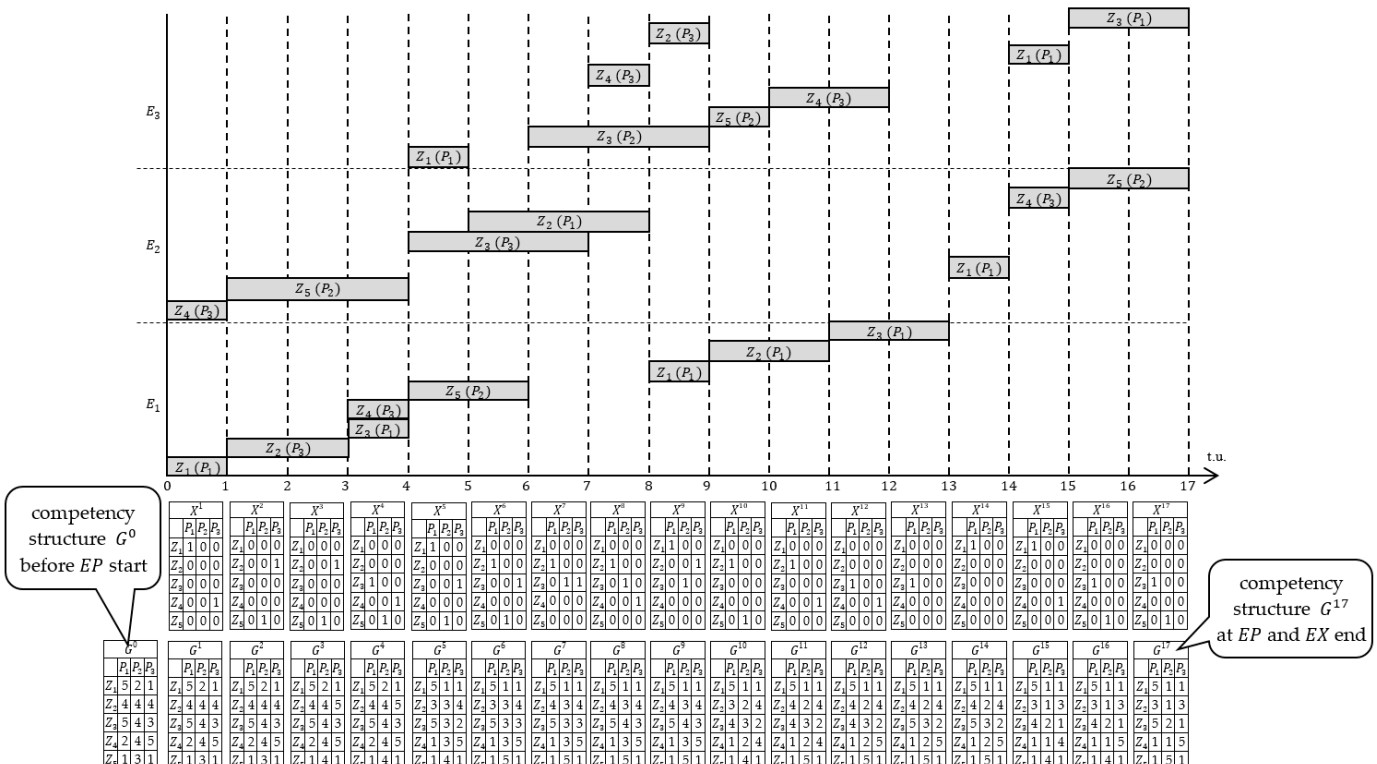

**Figure 9.** Solution to the synthesis problem in which the portfolio *EP* was executed according to the sequence of assignments $X^1$, $X^2$, ..., $X^{17}$ with task rotation but, however, with an additional *EX* order.

## 5. Computational Experiments for Scalability Assessment

In addition to qualitative experiments aimed at illustrating use cases of the proposed approach, quantitative experiments have also been carried out to assess the scale of solvable problems. For this purpose, an assessment of the problem-solving time of the analysis and synthesis was carried out for a different number: employees and tasks of projects that make up portfolio *E*. In the experiments, it was assumed that portfolio *E* consists of two *EP* set projects (2/3 of the number of tasks of portfolio *E*) and one additional *EX* project (1/3 of portfolio *E* tasks). To implement portfolio *E*, a team of employees with the following numbers is used: $K = 3, 4, 5, 6$. For each experiment (computer parameters: Intel Core i7-M4800MQ 2.7 GHz, 32 GB RAM), the expected levels of the competence structure *SG\** were arbitrarily assumed.

The conducted experiments allow for assessing the possibilities of searching for project portfolio implementation plans in situations of introducing new *EX* orders. The obtained results (see Table 6) demonstrate that the solution to analysis problems is always obtained in less than two seconds. The search for rotational plans (i.e., the synthesis problem) already requires significant computational outlays. Figure 10 shows that the time to solve the synthesis problem increases exponentially with the number of tasks of the project portfolio *E*.

The results of the experiments demonstrate that the proposed approach can be used to support decisions in the online mode (i.e., <10 min) in the synthesis of task rotation schedules for projects of no more than four employees and 27 tasks in portfolio *E*.

**Table 6.** Analysis and synthesis problem-solving times for different *E* project portfolios.

| Number of Employees *K* | Number of Project Portfolio *E* Tasks | The Expected Degree of Competence Structure *SG\** | Time Spent on Solving the Analysis Problem [s] | Time Spent on Solving the Synthesis Problem [s] |
|---|---|---|---|---|
| 3 | 15 | 45 | 1 | 16 |
| | 18 | 54 | 1 | 30 |
| | 21 | 63 | 1 | 57 |
| | 24 | 72 | 1 | 111 |
| | 27 | 81 | 1 | 240 |
| | 30 | 90 | 1 | 600 |
| 4 | 15 | 60 | 1 | 18 |
| | 18 | 72 | 1 | 41 |
| | 21 | 84 | 1 | 88 |
| | 24 | 96 | 1 | 190 |
| | 27 | 108 | 1 | 360 |
| | 30 | 120 | 1 | 880 |
| 5 | 15 | 60 | 1 | 55 |
| | 18 | 90 | 1 | 130 |
| | 21 | 105 | 1 | 280 |
| | 24 | 120 | 1 | 740 |
| | 27 | 135 | 2 | 1901 |
| | 30 | 150 | 2 | 3651 |
| 6 | 15 | 75 | 1 | 240 |
| | 18 | 108 | 2 | 440 |
| | 21 | 126 | 2 | 1256 |
| | 24 | 144 | 2 | 2589 |
| | 27 | 162 | 2 | 6987 |
| | 30 | 45 | 2 | >7000 |

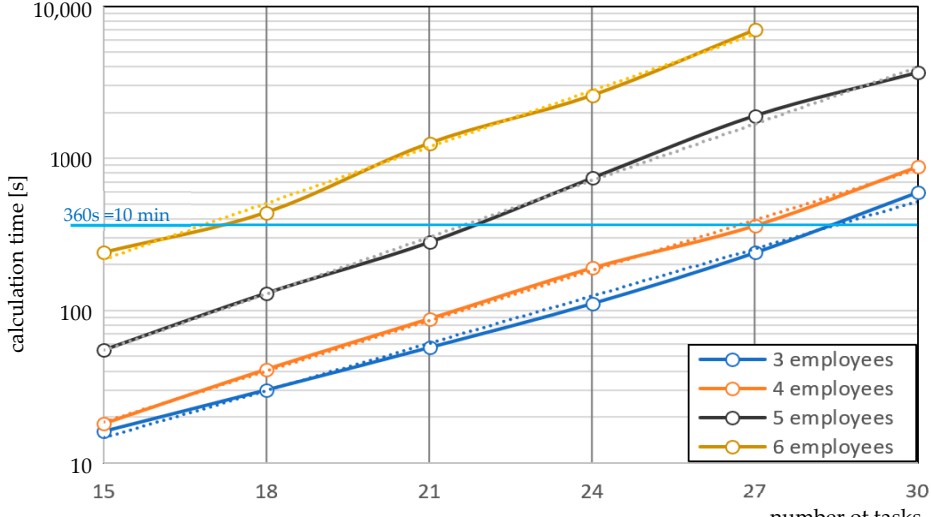

**Figure 10.** Time to determine the solution to the synthesis problem: results of experiments are presented by solid lines while trend functions are represented by dashed lines.

## 6. Conclusions

Supplementing the already implemented portfolio of projects with newly introduced ones changes the availability of the used human resources (work force), affecting its schedule and planned efficiency. Satisfying the related needs of dynamic (in online mode) resource scheduling imposes limits on their multi-purpose nature, including the expectations of minimizing resources' size, robustness (e.g., to employee absenteeism) as well as the need to refresh the multi-tasking competences of team members. In that context, the presented study attempts to develop an analytical model (implementing the declarative modeling paradigm) of a job rotation scheduling that allows for shaping the competence structure of the employed staff. The main assumption is that the level of the employee's skills is determined according to the curves of learning and forgetting, which allows for meeting the expectations of both skill enhancement and multi-skill cultivation. The case studies presented, implementing this model, illustrate its usefulness in planning staffing that guarantees a robust task rotation schedule (i.e., allowing for taking up a new production order without reducing the level of competence of the staff employed).

It should be noted that the presented approach can be used only for a project portfolio described by deterministic data. The use of stochastic models describing the random (i.e., taking into account the impact of the human factor) nature of the processes' conditioning the implementation of the project portfolio is not possible due to the difficulty of obtaining reliable random samples allowing for the identification of the density distribution of random variables and the synthesis of stochastic parameters of variables conditioning the expected implementation of the portfolio. However, directed fuzzy numbers can be used to describe imprecise variables, which are easily implemented in a constraints programming environment [46,47]. The possibility of using this type of representation will be the subject of our future research. In that context, the future work will be dedicated to extending the model to cases related to the uncertainty of operation times, planned order completion dates, and the skill level (expressed in fuzzy numbers) as well as the DSPS problem under a dynamic environment (taking into account the ad hoc leaving of some employees and the recruitment of new ones with, however, different competences). In a broader context, the question that dominates this study is: Can a given allocation of members of a multi-skilled team guarantee a job rotation schedule that can maintain the current level of its competences? This question will be expanded to include expectations related to minimizing the size of the team while maximizing its robustness.

**Author Contributions:** Conceptualization, E.S., G.B. and Z.B.; methodology, E.S. and G.B.; software, G.B.; validation, A.T.; formal analysis, G.B.; investigation, E.S.; resources, A.T.; data curation, E.S. and A.T.; writing—original draft preparation, Z.B.; writing—review and editing, Z.B. and E.S.; visualization, G.B. and E.S.; supervision, Z.B.; project administration, G.B. and E.S. All authors have read and agreed to the published version of the manuscript.

**Funding:** This research received no external funding.

**Institutional Review Board Statement:** Not applicable.

**Informed Consent Statement:** Not applicable.

**Data Availability Statement:** Not applicable.

**Conflicts of Interest:** The authors declare no conflict of interest.

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
