# Peer review of "Project Portfolio Planning Taking into Account the Effect of Loss of Competences of Project Team Members"

_applsci, doi:10.3390/app13127165_

Round 1

Reviewer 1 Report

Dear authors,

The paper is very interesting and the topic is very opportune in the context of Industry 5.0.

The document is well-structured and the proposed solution well-described.

As is characteristic of mathematical modelling, the incorporation of stochastic parameters becomes difficult and limiting. In this sens, it would be advisable to include a discussion section of results highlighting the main limits of the proposed solution (inclusion of stochastic task times, modifiable precedences, etc.).

It is also important to refer for who is directed this solution. Who is going to use this? It demands a DSS incorporating the model?

Congratulations for the work!

The quality of English language is good. Only minor editing required.

Author Response

The authors thank the reviewers for very helpful comments and suggestions. The authors have incorporated the reviewers’ comments in the revised manuscript.

 Responses to the reviewers’ comments

Reviewer #1: The paper is very interesting and the topic is very opportune in the context of Industry 5.0. The document is well-structured and the proposed solution well-described.

Remark 1: As is characteristic of mathematical modelling, the incorporation of stochastic parameters becomes difficult and limiting. In this sens, it would be advisable to include a discussion section of results highlighting the main limits of the proposed solution (inclusion of stochastic task times, modifiable precedences, etc.).

Response: Thank you for your comment. You are right that the direct use of stochastic parameters in the proposed model is not possible. Our research so far, however, has shown that imprecise parameters can be modeled using the representation of so-called directed fuzzy numbers. In this context, the Conclusions has been extended by the following paragraph.

Page 16. “It should be noted that the presented approach can be used only for a project portfolio described by deterministic data. The use of stochastic models describing the random (i.e., taking into account the impact of the human factor) nature of the processes conditioning the implementation of the project portfolio is not possible due to the difficulty of obtaining reliable random samples allowing the identification of the density distribution of random variables and the synthesis of stochastic parameters of variables conditioning the expected implementation of the portfolio. However, directed fuzzy numbers can be used to describe imprecise variables, which are easily implemented in a constraints programming environment [42, 43]. The possibility of using this type of representation will be the subject of our future research.”

[42]   Bocewicz, G.; Banaszak, Z.; Rudnik, K.; Smutnicki, C.; Witczak, M.; Wójcik, R. An ordered-fuzzy-numbers-driven approach to the milk-run routing and scheduling problem. Journal of Computational Science 2021, 49, 101288, https://doi.org/10.1016/j.jocs.2020.101288.

[43]   Rudnik, K.; Bocewicz, G.; KuciÅ„ska-Landwójtowicz, A.; Czabak-Górska, I.D. Ordered fuzzy WASPAS method for selection of improvement projects. Expert Systems with Applications 2021, 169, 114471, https://doi.org/10.1016/j.eswa.2020.114471.

Remark 2: It is also important to refer for who is directed this solution. Who is going to use this? It demands a DSS incorporating the model?

 Response: Thank you for your comment. The Introduction section has been expanded with the following paragraph:

 Page 3. “The proposed DMRPSP reference model can be implemented in any declarative programming environment (Gurobi, IBM ILOG, Lingo, etc.). The solver built in this way can complement the existing DSS system with project manager support to answer the following questions:

  1. What change in the level of employee competences will result in the acceptance of additional orders to the already adopted plan of project portfolio implementation?
  2. Is it possible to accept additional orders during the implementation of the planned portfolio of projects while maintaining the required level of employees’ competences?”

Reviewer 2 Report

Page 4, fourth paragraph, “They include stochastic models (aimed at e.g., risk assessment), operational research models (e.g., based on integer or mixed-integer programming, dynamic programming, etc.), simulation models (e.g., based on the multiagent concept), artificial intelligence models and fuzzy models (using e.g., population algorithms, metaheuristics, fuzzy logic algorithms, etc.).” Is it possible to give reference for each model as example?

What do the dashed line represent in the figure 10.

Page 14, “Figure 10 shows that the time to solve the synthesis problem increases exponentially with the number of tasks of the project portfolio ?.” But in figure 10, the line seems more like increasing linearly, instead of exponentially.

Page 6, “It is arbitrarily assumed that after the completion of the project portfolio, the degree of competence structure should be SG≥25 (i.e., around 25% of the maximum value).” How did the author determine the value of 25 for SG.

English Language can be improved.

Author Response

The authors thank the reviewers for very helpful comments and suggestions. The authors have incorporated the reviewers’ comments in the revised manuscript.

Reviewer #2:

Remark 1: Page 4, fourth paragraph, “They include stochastic models (aimed at e.g., risk assessment), operational research models (e.g., based on integer or mixed-integer programming, dynamic programming, etc.), simulation models (e.g., based on the multiagent concept), artificial intelligence models and fuzzy models (using e.g., population algorithms, metaheuristics, fuzzy logic algorithms, etc.).” Is it possible to give reference for each model as example?

 Response: Thank you for your comment. The following references have been added:

  1. Felberbauer, T.; Gutjahr, W.J.; Doerner, K.F. Stochastic project management: Multiple projects with multi-skilled human resources. Sched. 2019, 22, 271–288. https://doi.org/10.1007/s10951-018-0592-y.
  2. Hewitt, M.; Chacosky, A.; Grasman, S.E.; Thomas, B.W. Integer programming techniques for solving non-linear workforce planning models with learning. J. Oper. Res. 2015, 242, 942–950.
  3. Zheng, X; Wang, L. A multi-agent optimization algorithm for resource constrained project scheduling problem. Expert Syst. Appl. 2015, 42(15-16), 6039-6049. https://doi.org/10.1016/j.eswa.2015.04.009.
  4. Asensio-Cuesta, S.; Diego-Mas, J.A.; Canos-Daros, L.; Andres-Romano, C. A genetic algorithm for the design of job rotation schedules considering ergonomic and competence criteria. J. Adv. Manuf. Technol. 2012, 60, 1161–1174. https://doi.org/10.1007/s00170-011-3672-0.
  5. Jafari, H.; Haleh, H. Nurse scheduling problem by considering fuzzy modeling approach to treat uncertainty on nurses’ preferences for working shifts and weekends off. Optim. Ind. Eng. 2019. https://doi.org/10.22094/joie.2019.576759.159.

 Remark 2: What do the dashed line represent in the figure 10.  

Response: Thank you for pointing this out. The dashed line represents trend lines approximated as an exponential function. The figure caption has been changed to the following:

Page 15. “Time to determine the solution to the synthesis problem: results of experiments are presented by solid lines while trend functions are represented by dashed lines.”

Remark 3: Page 14, “Figure 10 shows that the time to solve the synthesis problem increases exponentially with the number of tasks of the project portfolio ?.” But in figure 10, the line seems more like increasing linearly, instead of exponentially.

 Response: Thank you for your comment. Figure 10 presents the functions of calculation time for the case of the synthesis problem on a logarithmic scale. For this kind of scale, the exponential functions look like linear functions.

 Remark 4: Page 6, “It is arbitrarily assumed that after the completion of the project portfolio, the degree of competence structure should be SG≥25 (i.e., around 25% of the maximum value).” How did the author determine the value of 25 for SG.

Response: Due to the didactic nature of the presented example, an arbitrary value was adopted (as input data). Our observation shows that typical level of degree of competence structure in practice is between 15%-60% (depends on the industry and the specifics of the project).

Reviewer 3 Report

Good day and complements on a good study

The study makes for an interesting read. I would like to suggest some improvements please.

1. the overall practical application of the study must be better highlighted

2. the method is not carved out into a methods sections, this also implies conformance to research processes. i.e. research grounding of the method.

3. The paper must have a theoretical grounding, with resonance in the method.

4. The declaration of variables comes to late in the paper, these should be earlier.

5. please provide a structured discuss on the interpretation of the results

Minor, please read fully

Author Response

The authors thank the reviewers for very helpful comments and suggestions. The authors have incorporated the reviewers’ comments in the revised manuscript.

Reviewer #3:

The study makes for an interesting read. I would like to suggest some improvements please.

Remark 1: The overall practical application of the study must be better highlighted.

Response: The issues and methods of job rotation are presented in many publications emphasizing the benefits resulting from it (e.g., related to boredom, burnout, distraction etc.) :

  • Sebt, V.; Ghasemi, S.S. Presenting a Comprehensive Smart Model of Job Rotation as a Corporate Social Responsibility to Improve Human Capital. IJSOM 2021, 8, 212–231, https://doi.org/10.22034/ijsom.2021.2.7
  • Asensio-Cuesta, S.; Diego-Mas, J.A.; Canos-Daros, L.; Andres-Romano, C. A genetic algorithm for the design of job rotation schedules considering ergonomic and competence criteria. Int. J. Adv. Manuf. Technol. 2012, 60, 1161–1174, https://doi.org/10.1007/s00170-011-3672-0
  • Gowsalya, R.S.; Jijo Francis, J. A Study on Employee Job Rotation. Int. J. Res. Trends Innov. 2017, 2, 205–210
  • Abiante, U.A.D. Impact of Teachers Job Rotation on Students’ Academic Performance in Rivers State. Niger. Int. J. Innov. Educ. Res. 2018, 6, 18–26
  • Cherotich, S.; Rop, W.; Bett, A. The Relationship between Job Rotation and Employee Performance in Level-Four Hospitals within the South Rift region in Kenya. Int. J. Sci. Res. Publ. 2021, 11, 139–145, https://doi.org/10.29322/IJSRP.11.09.2021.p117XX
  • Oparanma, A.O.; Nwaeke, I.L. Impact of Job Rotation on Organizational Performance. Journal of Economics. Manag. Trade 2015, 7, 183–187, https://doi.org/10.9734/BJEMT/2015/12051

The extension of job rotation problems to environments forcing the employment of multi-skilled employees, as is the case in services (education, medicine) or military (aviation, special units), implies another need for rotation: to maintain the potential of personnel in readiness. Examples of works in this field are illustrated in the following papers:

  • MaÅ‚achowski, B.; Korytkowski, P. Competence-based performance model of multi-skilled workers. Comput. Ind. Eng. 2016, 91, 165–177, https://doi.org/10.1016/j.cie.2015.11.018
  • Heimerl, C.; Kolisch, R. Work assignment to and qualification of multi-skilled human resources under knowledge depreciation and company skill level targets. Int. J. Prod. Res. 2010, 48, 3759–378
  • Felberbauer, T.; Gutjahr, W.J.; Doerner, K.F. Stochastic project management: Multiple projects with multi-skilled human resources. J. Sched. 2019, 22, 271–288, https://doi.org/10.1007/s10951-018-0592-y

Some of the research studies mentioned above, as well as the bibliographic sources referring to them, are presented in sections including the Introduction and Related works.

Remark 2: The method is not carved out into a methods sections, this also implies conformance to research processes. i.e. research grounding of the method.

 Response: The proposed approach is to introduce the data specified in the reference model, corresponding to one of two forms of the problem: analysis or synthesis. If we want to talk about the method in this context, one could consider alternating the approach. First, according to the scheme, is there an acceptable solution to the analysis problem for the input data entered? If so, stop. If not, answer the question: are there other inputs that result in an acceptable solution to the synthesis problem? To sum up, accepting your remark, the problem statement (see Section 3.3) has been extended by the following paragraph:

Page 10. “The proposed CS model (12) is part of an iterative (alternating) method of solving analysis and/or synthesis problems [41]. The essence of this method is a scheme of alternating approaches. To solve analysis problems, use the question:

  • Is there an admissible solution for the input data (competence structure with a given competence level)?

For synthesis problems use the question:

  • Are there other inputs (e.g., assignment of workers to tasks) resulting in an admissible solution (i.e., required level of competence structure)?

This method was used in the experiments described in the next section.”

[41] Szwarc, E.; Wikarek, J., Gola, A.; Bocewicz, G.; Banaszak, Z. Interactive Planning of Competency-Driven University Teaching Staff Allocation. Applied Sciences-Basel 2020, 10(14), https://doi.org/10.3390/app10144894.

Remark 3: The paper must have a theoretical grounding, with resonance in the method.

Response: The discussion of the problem under consideration in the context of the related theoretical foundations is presented in Related Works (see Section 2). In addition, in order to emphasize its specificity determining the existence of acceptable solutions, an illustrative example has been introduced (see Section 3.1.). The method of solving the problem formulated in Section 3.3. (see the answer to Remark 2) should be sought in the alternating solving of analysis and synthesis problems. In other words, alternating a search for an answer to the questions of whether an admissible solution is reachable and what changes, and what parameters guarantee this kind of solution.

 Remark 4: The declaration of variables comes to late in the paper, these should be earlier.

 Response: The meaning of all variables in the reference model was described earlier in the illustrative example (see Section 3.1), in order to facilitate the transmission of intuitions behind their formal (related to the assigned designations) relationships and constraints presented later. In other words, introducing a list of variables earlier without accompanying constraints, risks losing the intuition associated with them. Therefore, it seems that a better narrative would be the one currently adopted.

Remark 5: Please provide a structured discuss on the interpretation of the results.

 Response: Please note that the discussion of the results obtained, including the effectiveness of the implemented model, is described in the Case study (see Section 4) and in Computational experiments (see Section 5). In addition, the Conclusions (see Section 6) presents the main advantages and limitations of the proposed solution.
